# Preparation and Properties of Organically Modified Na-Montmorillonite

**DOI:** 10.3390/ma16083184

**Published:** 2023-04-18

**Authors:** Yan Qian, Zeen Huang, Guantao Zhou, Chenan Chen, Yuhang Sang, Zuolong Yu, Legao Jiang, Yuning Mei, Yunxiao Wei

**Affiliations:** 1Biology and Environment Engineering College, Zhejiang Shuren University, Hangzhou 310015, China; 2Jiangsu Province Key Laboratory of Fine Petrochemical Engineering, Changzhou University, Changzhou 213164, China; 3Zhejiang Hongyu New Materials Co., Ltd., Huzhou 313113, China

**Keywords:** montmorillonite, organic modification, characterization, property

## Abstract

This study investigates the montmorillonite (MMT) content, rotational viscosity, and colloidal index of sodium montmorillonite (Na-MMT) as a function of the sodium agent dosage, reaction time, reaction temperature, and stirring time. Na-MMT was modified using different octadecyl trimethyl ammonium chloride (OTAC) dosages under optimal sodification conditions. The organically modified MMT products were characterized via infrared spectroscopy, X-ray diffraction, thermogravimetric analysis, and scanning electron microscopy. The results show that the Na-MMT with good properties (i.e., the maximum rotational viscosity and highest Na-MMT content with no decrease in the colloid index) was obtained at a 2.8% sodium carbonate dosage (measured based on the MMT mass), a temperature of 25 °C, and a reaction time of two hours. Upon organic modification of the optimized Na-MMT, OTAC entered the NA-MMT interlayer, and the contact angle was increased from 20.0° to 61.4°, the layer spacing was increased from 1.58 to 2.47 nm, and the thermal stability was conspicuously increased. Thus, MMT and Na-MMT were modified by the OTAC modifier.

## 1. Introduction

Montmorillonite (MMT) is the main component of bentonite, which is composed of aluminosilicate silica tetrahedron and alumina octahedron. Calcium-based MMT is the main type of MMT found in China [1,2]. The loose-layered structure of MMT makes it highly dispersible, adsorbent, and expandable, such that it is easily activated and modified [3,4,5]. Furthermore, due to these properties, MMTs are widely used in environmental adsorption as industrial fillers, medicinal carriers, and food additives [6,7,8,9].

Owing to its poor quality and unstable performance, calcium-based MMT does not meet the requirements for direct applications. Sodium modification of crude MMT is a prerequisite for the purification and further modifications of MMT. Sodium-based and lithium-based modifications of MMT can increase the MMT content in bentonite, which is useful for improving its application value [10]. Using surfactants, organics with different chain lengths are inserted into the MMT layers, which considerably increases the layer spacing and enhances the dispersion of MMT in organic solvents [11]. Replacing the calcium and sodium ions between the MMT layers with metal cations considerably improves the adsorption properties of F^−^ and Cr^2+^, which is advantageous for depolluting water [12,13,14]. In addition to being used for the adsorption of heavy metal ions and serving as a plugging agent for drilling fluids, MMT can be used as a packaging film component to change the barrier property and subsequently be applied in food packaging [15,16]. MMT can also be blended with coating materials and coat the surfaces of fruits and vegetables to extend their shelf life [17,18,19].

The organic modification of MMT involves the replacement of exchangeable cations between the MMT layers with cationic surfactants (such as organic amines and quaternary ammonium salts) and nonionic surfactants. Organic groups are thus inserted between the MMT layers to change the hydrophilicity of its surface so as to produce lipophilic organic MMT that can be dispersed in organic solvents [5]. Given the different sources of MMTs, their compositions can be broadly tuned, making it difficult to determine the optimal conditions for preparing organic MMT, such as the appropriate reaction medium, temperature, and pH. The effectiveness of organic modifiers, such as cetyltrimethyl bromide and octadecyltrimethyl ammonium chloride (OTAC), depends on the preparation parameters. Among the organic modifiers, quaternary ammonium salts are the best organic modifiers, particularly the double-chain, long-chain fatty acid quaternary ammonium salts [11]. Based on the layer spacing of organic MMT, its gelling property in dispersants and its shearing ability can be directly used to determine different optimal formulations [11]. This study used raw MMT (Zhejiang Hongyu New Material Co., Ltd., Huzhou, China) to investigate the influences of different sodium conditions on the resulting product. Moreover, the intercalation capacity and product layer spacing were compared as a function of intercalation with different fractional contents of organic compounds.

## 2. Materials and Methods

### 2.1. Materials

Raw MMT was obtained from Zhejiang Hongyu New Materials Co., Ltd. (Huzhou, China). OTAC was purchased from Aladdin (Shanghai Aladdin Biochemical Technology Co., Ltd., Shanghai, China). Sodium carbonate and all other chemicals were purchased from Shanghai Lingfeng Chemical, Ltd. (Shanghai, China). All materials were used as received.

### 2.2. Preparation of Sodium Montmorillonite (Na-MMT)

First, 20 g of raw MMT with a mass ratio of 52.27% was added to 200 mL of deionized water in a 500 mL three-necked flask. The raw MMT and deionized water were uniformly mixed via stirring to form a slurry. Subsequently, various weights of solid sodium carbonate (measured based on the mass ratio of crude MMT, g/g) and 200 mL of deionized water were added to the flask and stirred continuously. The sodium slurry was then poured into a centrifuge tube, stirred, and centrifuged at 2000 rpm for two minutes. The resultant precipitate was dried at 105°C and subsequently crushed using a 200-mesh sieve (0.074 µm) to obtain the Na-MMT. The preparation of the Na-MMT was based on company experience.

### 2.3. Organic Modification of Na-MMT

The Na-MMT prepared under the optimized conditions was mixed into a 30% slurry (g/mL by volume of aqueous solution), stirred at 500 rpm, heated to 70 °C in a water bath, and stirred for one hour in the water bath. The pH was adjusted to 4.0 by adding diluted sulfuric acid. OTAC (organic intercalation agent) at mass ratios of 30, 50, and 80% (g/g, based on the mass ratio of Na-MMT) was slowly added into the slurry, and it stirred for two hours to obtain the organically modified MMT (referred to as OMMT-1, OMMT-2, and OMMT-3, respectively). The resultant slurries with organic intercalation were centrifuged for two minutes at 2000 rpm. The precipitates were dried at 105 °C and crushed using a 200-mesh sieve.

### 2.4. Analysis and Characterization

#### 2.4.1. Sample Analysis

Measurement of blue absorption: Na-MMT samples were dispersed in an aqueous solution and titrated with a methylene blue standard solution. During the titration, the amount and time interval were controlled, and a drop of test liquid was dropped onto filter paper using a glass rod until a light green halo ring appeared around the central dark blue spot. At this point, the titration was stopped. The volume of the methylene blue standard solution for the titration was recorded, and the MMT content was calculated as follows [20]:(1)$$M\left(\%\right)=\frac{CV}{4.42\;m}\times 100\%,$$
where *M* denotes the Na-MMT content (%), *C* denotes the concentration of the methylene blue standard solution (mol/L), *V* denotes the volume of the methylene blue solution consumed via titration (mL), 4.42 is the conversion factor, and *m* denotes the sample mass (g).

Measurement of rotary viscosity: A 5 g sample of Na-MMT and 45 mL of xylene were weighed and placed into a beaker and stirred for five minutes to form a viscous colloid. The colloid was then poured into the test container of a rotary viscometer to measure the rotational viscosity. Data were recorded once the rotary viscometer pointer became stable [21].

Measurement of colloidal rate: An appropriate amount of xylene and 1.5 g of the sample were poured into a 100 mL plug-type measuring cylinder and shaken for five minutes. Subsequently, xylene was added to the 100 mL cylinder. The shaking continued until the sample was evenly dispersed. Then, the mixture was left undisturbed for two hours. The volume of free xylene in the upper layer of the cylinder was recorded. The colloidal rate was calculated using Equation (2) [22]:(2)$$W\left(\%\right)=\frac{100-V}{100}\times 100\%,$$
where *W* denotes the colloidal rate (%), and *V* denotes the upper free volume of the measuring cylinder (mL).

#### 2.4.2. Sample Characterization

The chemical states of the samples were analyzed using a PHI 5000 X-ray photoelectron spectroscopy (XPS) system from the Ulvac-Phi company (Nagasaki, Japan) equipped with a monochromatic Al K-alpha source. The samples were analyzed via Fourier transform infrared (FTIR) spectroscopy using an iS50 infrared spectrum analyzer from NICOLET with a resolution of 4 cm^−1^ over the wavenumber range of ~500–4000 cm^−1^. The samples were scanned 32 times. A Japan Shimadzu 6100 X-ray powder diffractometer (XRD) was used for the XRD analysis. The X-ray tube voltage and current were 40 kV and 30 mA, respectively. Raw and modified MMT samples (10 mg) were separately mixed with a KBr tablet for the XRD analysis over a scanning range of ~1.5°–40° in steps of 0.2°. The thermogravimetric analysis (TGA) was performed using a DTG-60 thermogravimetric instrument from the Shimadzu Company (Kyoto, Japan). First, 5 mg of a sample was measured in a nitrogen atmosphere using a nitrogen flow of 20 mL/min. The temperature was raised from room temperature to 1000 °C at a heating rate of 20 °C/min. An S-570 scanning electron microscope (SEM) from Hitachi (Japan) was used to analyze the sample morphology. The static contact angle measuring instrument JC2000D3, produced by Shanghai Zhongchen Digital Technology Equipment Co., Ltd. (Shanghai, China), was used to observe the sample contact angle. A 1 g sample was pressed under 18 MPa and placed in the observation room of the measuring instrument. Water drops were added, and photographs were captured five seconds later to determine the contact angle.

## 3. Results and Discussion

### 3.1. Na-MMT Optimization Conditions

Sodification is a common method for purifying MMT [23,24]. Figure 1 shows the effect of the added sodium carbonate amount on the MMT content of the crude MMT samples. The MMT content and rotational viscosity gradually decreased with the increasing sodium dosage, whereas the colloid rate changed slightly. Since sodification replaces calcium ions in the MMT, it increases MMT dispersion in water, improves cation exchange capacity, and enhances thermal stability [25]. A sodium dosage of 2.8% was considered optimal because it resulted in the maximum rotational viscosity and highest Na-MMT content with no decrease in the colloid index (Figure 1).

Figure 2 shows the MMT properties as a function of the sodification time for the raw MMT samples. The sodification reaction was determined to be optimal at two hours. Upon increasing the reaction time, the MMT content in the suspension decreased, and the rotational viscosity decreased considerably. The decrease in the MMT content and rotational viscosity was caused by the dispersion of impurities from the other clays, such as kaolinite, attapulgite, and nonclay minerals, with long reaction times [26]. Therefore, the optimal reaction time for our experiment was two hours.

Figure 3 shows the MMT content, rotational viscosity, and colloidal index properties of Na-MMT as a function of the sodification temperature. The MMT content gradually increased with the increasing sodification temperature, whereas the rotational viscosity gradually decreased. Since the increase in temperature assists MMT dispersion in the solvent and expands the ion exchange area, a higher temperature is conducive to sodification. However, the increase in temperature also destroys the layered structure and causes a considerable decline in rotational viscosity. Therefore, the optimal sodification temperature was 25 °C.

Figure 4 shows the MMT properties as a function of the sodification stirring speed. When the stirring speed reached a certain value, it had a marginal influence on the MMT content after sodification. The MMT content exceeded 100% for stirring speeds of 300 and 400 rpm, while the rotational viscosity decreased considerably at 400 rpm. This decrease in rotational viscosity can be attributed to accelerated shear, which destroys the layered structure of the MMT [27]. However, the colloid rate was marginally affected by the changing stirring speed. Therefore, the rotational speed can be adjusted to suit the demands placed on the resultant Na-MMT.

Table 1 shows the elemental analysis results of the raw MMT and Na-MMT. The sodium content in Na-MMT was clearly higher than that in the raw MMT. Additionally, as shown in Table 2, the interlayer water was also partially replaced by other ingredients. XPS characterization indicated that sodification considerably modified the MMT.

### 3.2. Characterization of the MMT

Figure 5 shows the FTIR spectra of the MMT, Na-MMT, and three types of OMMT. For the MMT, stretching vibration absorption peaks appeared at 3626, 1640, 1045, 723, 520, and 460 cm^−1^ [28]. Upon sodification, the carbonate absorption peak of sodium carbonate appeared at 1430 cm^−1^ and then disappeared upon organic modification. The infrared absorption peaks of the OTAC also appeared in the IR spectra: –CH_2_–peaks appeared at 2918, 2873, and 1436 cm^−1^. The IR peak intensities gradually increased with the increasing amounts of intercalation agent. Additionally, the H_2_O absorption peaks of the organically modified MMT at 3450 and 1640 cm^−1^ were considerably weakened upon adding an intercalation agent. This weakening gradually increased with the increasing amount of intercalation agent. Furthermore, the intensities of the absorption peaks of Si–O and Al–O at 1045, 520, and 460 cm^−1^ were significantly enhanced, indicating that the bound water of the MMT was replaced, and other impurities were removed.

Figure 6 shows the contact angles of the different samples. The hydrophilicity of the raw MMT, Na-MMT, and OMMTs differed significantly upon modification. Although the MMT was treated with sodium, the contact angle of the Na-MMT changed only marginally, and the MMT remained strongly hydrophilic after sodification. Organic modification caused a conspicuous increase in the hydrophobicities of the OMMT samples, indicating that OTAC was adsorbed onto the surface or inserted into the MMT layers via other forms of attachment [29].

XRD characterization provides details of the atomic and molecular layer spacing of a sample. The layer spacing d_001_ effectively expresses the ability of an intercalation agent to intercalate between the MMT layers [30]. Figure 7 shows the XRD patterns and d_001_ values of all the samples. The intercalation interval of the organic products of OMMT-3 (d_001_ = 2.47 nm) was significantly longer than that of the raw material (MMT: d_001_ = 1.58 nm; Na-MMT: d_001_ = 1.54 nm). The fraction of OTAC entering the MMT increased with the increasing OTAC amount, although only slightly. The peak at a 2θ value of 27° was due to the small amount of zeolite and quartz in the sample, and its intensity gradually decreased with the increasing concentration of intercalation agent. This result indicates that the ability of the intercalation agent to enter the MMT interlayer is related to the type of molecule, the solvent condition, and the processing method [31].

Thermogravimetric analysis (TGA) can be used to quantitatively analyze the amount of intercalation agent inserted into the MMT and the thermal stability of the resulting OMMT. The TGA and derivative thermogravimetry results for the different samples are shown in Figure 8 and Figure 9, respectively, and Table 2. Figure 9 shows that the MMT and Na-MMT exhibited only two thermogravimetric stages: (1) from room temperature to 200 °C, which was caused by the volatilization of free water and interlayer water on the aluminosilicate surface, and (2) from ~600 to 800 °C, which was caused by MMT dehydroxylation and heat absorption [29]. However, owing to the different forms of adhesion and intercalation, the weight loss of OMMT caused by the volatilization and decomposition of the organic matter occurred at ~200–500 °C. Since the boiling point of free OTAC is 249 °C, TGA and DTG reveal several peaks near 250 °C corresponding to the surface, interlayer, and inner layers of the MMT. The weight loss rates of the OMMT–1, OMMT–2, and OMMT–3 in this stage were 15.43%, 17.46%, and 22.48%, respectively, which were caused by the differences in the amount of intercalation agent. The difference in the intercalation agent content between the obtained products became marginal with the increase in organic OTAC. This can primarily be attributed to the fact that the intercalation reaction ability is affected by many factors, such as the physical and chemical properties of the intercalation agent and the reaction conditions [11].

Figure 10 shows the SEM images of the different samples. MMT and Na-MMT exhibit large pore spacing with dark lamellar edges and weak dispersion. After organic modification, the lamellar edges became bright, with many particles attached to the surface, and fragmentation of the lamellar regions was conspicuous. These results show that the MMT was effectively modified by the modifier and that the modifier entered into the interlayer region and adhered to the MMT surface.

## 4. Conclusions

Sodium carbonate was used as the sodification agent of Ca–based MMT, and Na-MMT was prepared under the optimal sodification conditions determined here. Subsequently, the optimized Na-MMT was organically intercalated with varying amounts of OTAC. The sodification results indicate that the amount of sodification agent and the reaction temperature are the main factors affecting MMT sodification, followed by the reaction time and stirring rate. In the organically modified MMT, the organic intercalation agent could enter the MMT interlayer (intercalation mass fraction ≈ 15.43–22.48%), showing good hydrophobicity. The intercalation ability improved slightly with the increasing amount of intercalation agent.

## Figures and Tables

**Figure 1 materials-16-03184-f001:**
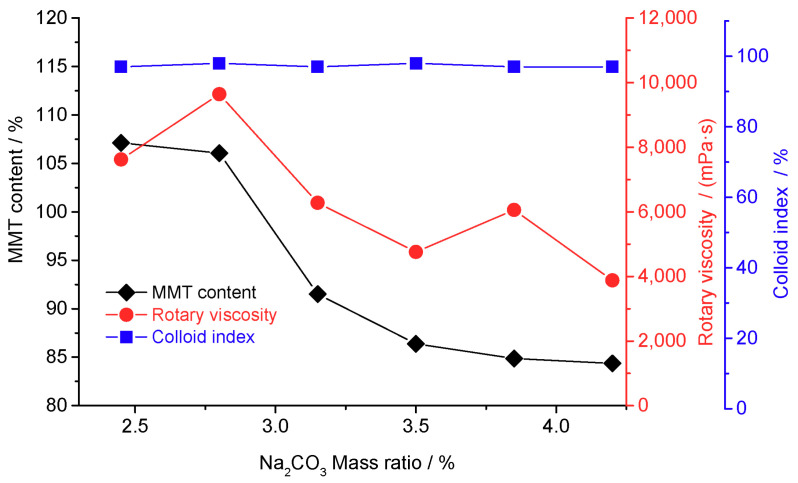
Properties of the montmorillonite (MMT) as a function of sodium carbonate addition (reaction conditions: 2 h, 25 °C, and stirring speed = 300 rev/min).

**Figure 2 materials-16-03184-f002:**
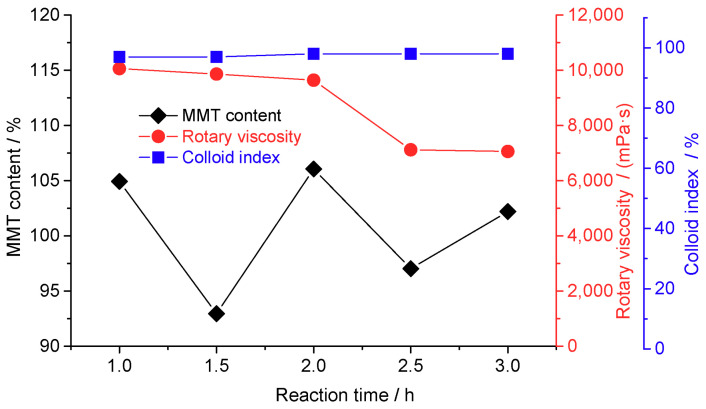
Properties of montmorillonite (MMT) as a function of the reaction time (reaction conditions: 25 °C, stirring speed = 300 rev/min, and sodium carbonate mass fraction = 2.8%).

**Figure 3 materials-16-03184-f003:**
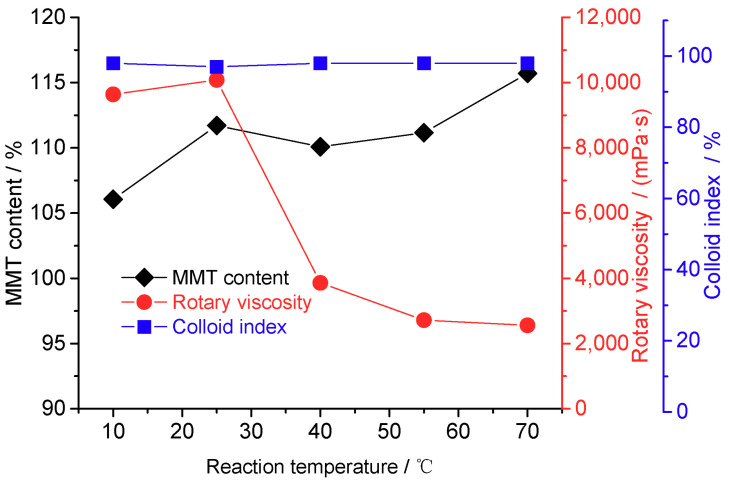
Properties of the montmorillonite (MMT) as a function of the reaction temperature (reaction conditions: sodium carbonate mass = 2.8%, 2 h, and stirring speed = 300 rev/min).

**Figure 4 materials-16-03184-f004:**
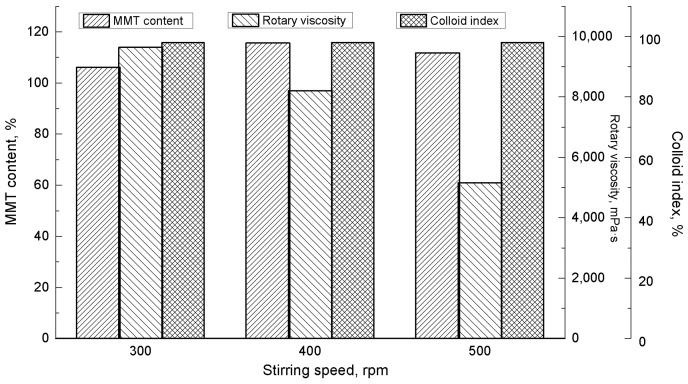
Properties of the montmorillonite (MMT) as a function of the stirring speed (reaction conditions: sodium carbonate mass = 2.8%, 2 h, and 25 °C).

**Figure 5 materials-16-03184-f005:**
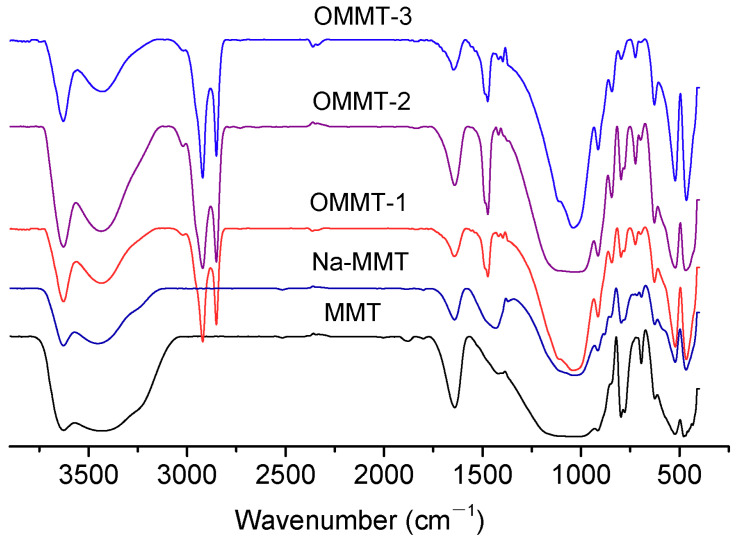
Fourier transform infrared (FTIR) spectra of the montmorillonite (MMT), Na-MMT, OMMT-1, OMMT-2, and OMMT-3 samples.

**Figure 6 materials-16-03184-f006:**
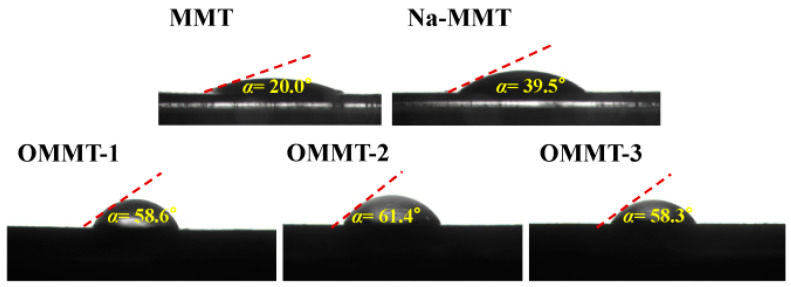
Photographs and corresponding contact angles of the montmorillonite (MMT), Na-MMT, OMMT-1, OMMT-2, and OMMT-3 samples.

**Figure 7 materials-16-03184-f007:**
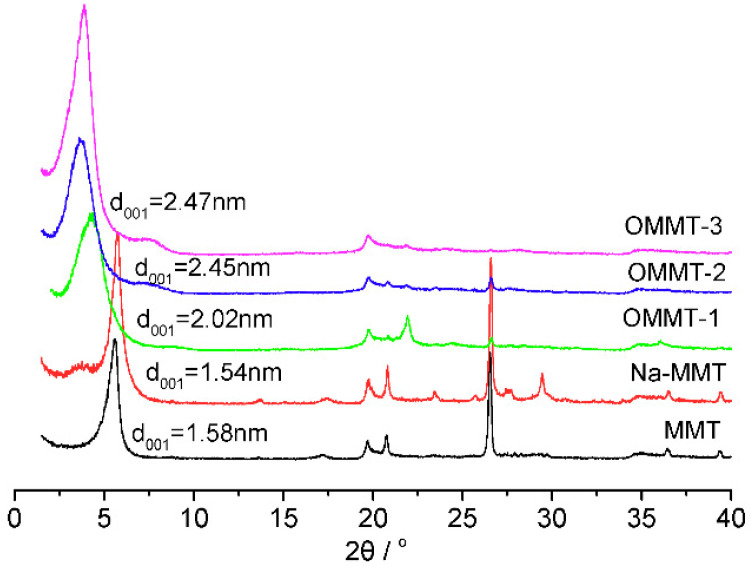
XRD patterns of the montmorillonite (MMT), Na-MMT, OMMT-1, OMMT-2, and OMMT-3 samples.

**Figure 8 materials-16-03184-f008:**
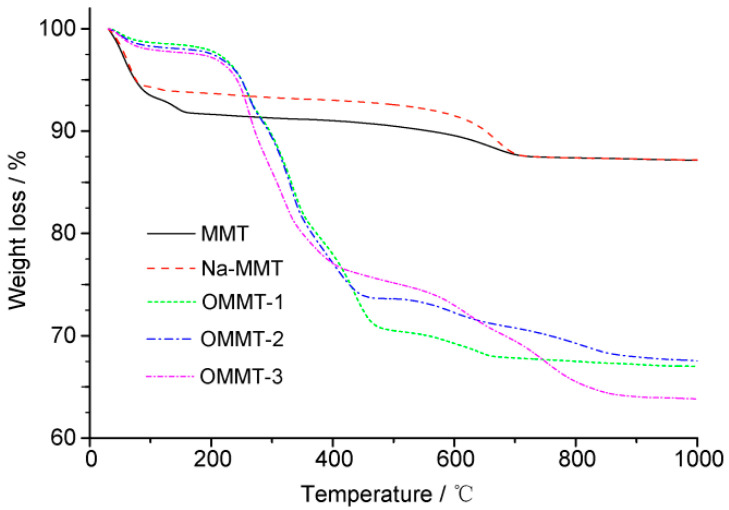
Thermogravimetric analysis (TGA) curves of the montmorillonite (MMT), Na-MMT, OMMT-1, OMMT-2, and OMMT-3 samples.

**Figure 9 materials-16-03184-f009:**
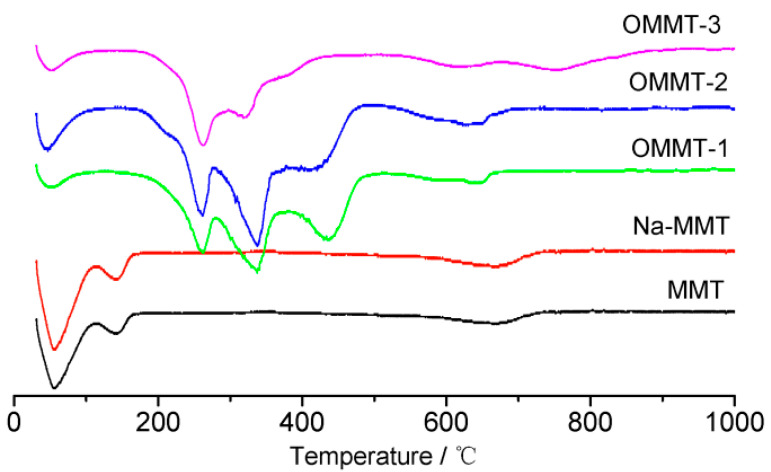
Derivative thermogravimetry curves of the montmorillonite (MMT), Na-MMT, OMMT-1, OMMT-2, and OMMT-3 samples.

**Figure 10 materials-16-03184-f010:**
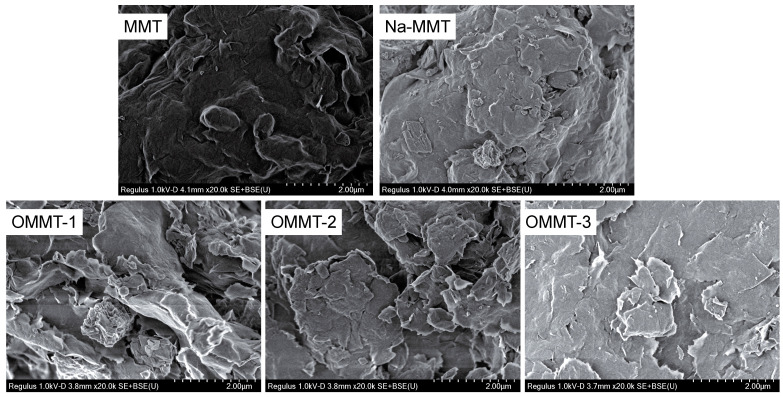
Scanning electron microscope (SEM) images of the montmorillonite (MMT), Na-MMT, OMMT-1, OMMT-2, and OMMT-3 samples.

**Table 1 materials-16-03184-t001:** Chemical composition of the (a) MMT and (b) Na-MMT.

Ingredient		SiO_2_	Al_2_O_3_	Fe_2_O_3_	MgO	Na_2_O	CaO	Other
Content/wt%	a	60.41	23.47	0.28	3.02	0.17	1.12	11.53
	b	61.06	22.51	0.23	3.55	4.36	0.76	7.53

**Table 2 materials-16-03184-t002:** Comparison of weight loss between the montmorillonite (MMT) and its products.

Sample	Weight Loss at 80–105 °C (%)	Weight Loss at 200–400 °C (%)	Weight Loss at 500–850 °C (%)	Residue (%)
MMT	6.85	5.97		87.18
Na-MMT	5.76	7.11		87.23
OMMT-1	2.56	15.43	12.52	69.49
OMMT-2	2.48	17.46	10.20	69.86
OMMT-3	2.29	22.48	11.38	63.85

## Data Availability

All data are freely available.

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
