# Peer review of "Preparation and Properties of Organically Modified Na-Montmorillonite"

_materials, 2023, doi:10.3390/ma16083184_

Round 1

Reviewer 1 Report

The manuscript materials-2260524 "Preparation and Properties of Organically Modified Montmorillonite" is a fundamental study of the interaction of smectite with octadecyl trimethyl ammonium chloride to improve the sorption properties of the mineral. The undoubted merit of the authors is a multidisciplinary approach and the use of various analytical skills in their work. However, there are a few areas where the authors can strengthen the manuscript. These are not related to the science essentially, but the presentation of their work in the text and the clarity of the findings they are trying to convey. These changes are major for careful tweaking, improving and extending a few points of the manuscript. Nevertheless, I think that the manuscript would be a valuable contribution to the Material.

1. Introduction is recommended to improve and emphasise the scientific novelty of the study.

2. In the introduction, please provide the nomenclature (according to IMA) definition of montmorillonite.

3. It is necessary to characterise the original montmorillonite. What is the chemical composition of the original montmorillonite? Than montmorillonite Zhejiang Hongyu New Materials Co. different from other smectites?

4. Section 2.4.2 Sample characterization. What amount was expanding smectite layers in the original mineral and after preparation by XRD scanning of oriented preparations determined?

5. Figures 1-3. The authors are recommended to clarify in the description of the figures why the amount of montmorillonite exceeds 100% along the y-axis. Besides, the signatures are too similar. In this case, the authors should combine these figures into one.

6. Please try to avoid interpretations and discussions with references in the section "Results".

7. Figure 5. Interpretation of peaks with value labels is required to improve perception. Is it absorption and transmission?

8. Section 3.2 Characterization of MMT properties. "With increasing amounts of intercalation agent, the IR peak intensities gradually increased" is a speculative claim. How to explain it? These peaks simply increase as the proportion of the STAC agent increases.

9. Please explain in detail the following statement "is related to the type of molecule, the solvent condition, and the process method". What types of structures and conditions do authors discuss?

10. I highly recommend adding a "Discussions" section.

11. In Discussions, consideration of the obtained results with world data should be expanded to increase the scientific significance of the article. In other words, the authors should justify their conclusions by comparing their results with others' works, emphasising the similarities and differences between their findings and those works. In general, this problem is present throughout the discussion chapter. Furthermore, the authors should give more credit to others' works and discuss their findings in the context of previous work.

Author Response

  1. Introduction is recommended to improve and emphasise the scientific novelty of the study.

Thank you for your comment. I think that the scientific novelty of the study is to modify bentonite from specific sources. Although many articles discuss the sodiation and organization of MMT, the relevant data are inconsistent. This study improves this limitation.

  1. In the introduction, please provide the nomenclature (according to IMA) definition of montmorillonite.

Thank you for the comment. I found the nomenclature at http://cnmnc.units.it/. The nomenclature is montmorillonite, and the formula is (Na,Ca)0.3(Al,Mg)2Si4O10(OH)2·nH2O. However, I think the definition is unnecessary, and it is provided at https://link.springer.com/article/10.1007/s00269-007-0197-z.

  1. It is necessary to characterise the original montmorillonite. What is the chemical composition of the original montmorillonite? Than montmorillonite Zhejiang Hongyu New Materials Co. different from other smectites?

Thank you for the comment. I have added the results of X-ray photoelectron spectroscopy (XPS) characterization. The MMT from the aforementioned company is different from the others. The goal of the XPS experiment is to clarify this difference.

  1. Section 2.4.2 Sample characterization. What amount was expanding smectite layers in the original mineral and after preparation by XRD scanning of oriented preparations determined?

Thank you for the comment. However, unfortunately, I do not understand the meaning of your question. Sodiation essentially does not modify the smectite layers. Nevertheless, the XPS and TGA results reveal a slight difference in the smectite layers.  

  1. Figures 1-3. The authors are recommended to clarify in the description of the figures why the amount of montmorillonite exceeds 100% along the y-axis.

Thank you for the comment. I noticed the abovementioned observation (i.e., the amount of montmorillonite exceeds 100% along the y-axis)  when I was collating the data. The method of blue absorption is not accurate, which could be the reason for the abovementioned observation. Currently, there is no better method. However, the results show that the modification is considerably good.

Besides, the signatures are too similar.

Yes, because they are English names. However, I assure you these are the signatures of all authors.

In this case, the authors should combine these figures into one.

In each experiment, the amount of MMT differs, but the colloid index remains almost the same. In this solvent system, the colloid index is the reference value; therefore, we cannot combine the figures.

  1. Please try to avoid interpretations and discussions with references in the section "Results".

Thank you for the comment. I followed the Microsoft Word template and revised the section “Results” to the section “Results and Discussion.” 

  1. Figure 5. Interpretation of peaks with value labels is required to improve perception. Is it absorption and transmission?

Thank you for the comment. The text has been revised.

  1. Section 3.2 Characterization of MMT properties. "With increasing amounts of intercalation agent, the IR peak intensities gradually increased" is a speculative claim. How to explain it? These peaks simply increase as the proportion of the STAC agent increases.

Thank you for the comment. Note that STAC can be dissolved in hot water. After intercalation, the redundant STAC remains in the solvent; therefore, only the STAC intercalated and attached to the MMT surface is characterized. The TGA characterization is also an explanation. 

  1. Please explain in detail the following statement "is related to the type of molecule, the solvent condition, and the process method". What types of structures and conditions do authors discuss?

Thank you for the comment. As an organic intercalator, the quaternary ammonium salt is the best because the long-chain fatty acids in this salt are more effective than short-chain fatty acids. Additionally, double chains work better than single chains. Previous studies explain in detail the solvent conditions and processing method.

  1. I highly recommend adding a "Discussions" section.

Thank you for the comment. The “Results” section has been revised to “Results and Discussion.”

 In Discussions, consideration of the obtained results with world data should be expanded to increase the scientific significance of the article. In other words, the authors should justify their conclusions by comparing their results with others' works, emphasising the similarities and differences between their findings and those works. In general, this problem is present throughout the discussion chapter. Furthermore, the authors should give more credit to others' works and discuss their findings in the context of previous work.

Thank you for the constructive advice.

Reviewer 2 Report

The present paper reported by Qian Yan. Et.al. studied the modification of montmorillonite clay by Octadecyltrimethyl ammonium chloride agent. The structure of as-modified MMT was investigated using different characterization methods. I am providing some comments that need to be addressed. Overall, this paper can be acceptable for publication in the Journal of Materials after addressing the following point:

1-     Add reference peak list for in the XRD pattern and specify the (hkl) in Fig. 1. Identify the shift for peak positions.

2-     Use the Williamson-hall formula for calculation of grain size in the XRD analysis and compare with results of Scherrer formula. Explain the macrostrain with Williamson-hall formula.

1-     Identified the BET isotherms. According to the IUPAC classification, the adsorption-desorption BET isotherm should be specified. Also, the BJH plot is inquired.

2-     Clarify the formation mechanism of with more details in the synthesis procedure step by step.

3-     Introduction section should be modified and reflect clear enough background and more explanation about clay minerals and its applications. (Cite Ref: - The Journal of Physical Chemistry C 122 (29), 16498-16509 , 2018. - Scientific Reports 12 (1), 8103, 20022. - Fuel 320, 123933, 2022. -Journal of Elastomers & Plastics 53 (3), 241-257, 2021. - Journal of The Electrochemical Society 167 (2), 020544, 2020.)

4-     Correlate between XRD result and HR-TEM.

5-     The application of modified clay should be studied. 

Author Response

1-     Add reference peak list for in the XRD pattern and specify the (hkl) in Fig. 1. Identify the shift for peak positions.

Thank you for the comment. Unfortunately, I do not understand the meaning of the abovementioned comment. I have revised the content of the XRD pattern, which appears in Fig. 7.

2-     Use the Williamson-hall formula for calculation of grain size in the XRD analysis and compare with results of Scherrer formula. Explain the macrostrain with Williamson-hall formula.

Thank you for the good advice. However, I have used only the 2θ value of d001 to compare the effect of the intercalation; therefore, I cannot compare the results of the analysis based Williamson–Hall formula with those of the analysis based on the Scherrer formula.

1-     Identified the BET isotherms. According to the IUPAC classification, the adsorption-desorption BET isotherm should be specified. Also, the BJH plot is inquired.

Thank you for the excellent suggestion. However, this experiment is not discussed in the manuscript. I understand that the BET isotherms constitute solid evidence of the MMT structure before and after modification. However, the characterization presented in the manuscript also constitutes solid evidence of the same. 

2-     Clarify the formation mechanism of with more details in the synthesis procedure step by step.

The formation mechanism is detailed in Ref. [11]. The manuscript only identifies the difference between local bentonite and organically modified bentonite. 

3-     Introduction section should be modified and reflect clear enough background and more explanation about clay minerals and its applications. (Cite Ref: - The Journal of Physical Chemistry C 122 (29), 16498-16509 , 2018. - Scientific Reports 12 (1), 8103, 20022. - Fuel 320, 123933, 2022. -Journal of Elastomers & Plastics 53 (3), 241-257, 2021. - Journal of The Electrochemical Society 167 (2), 020544, 2020.)

Thank you for the references. I will cite these articles in my next manuscript: application of organically modified MMT.

4-     Correlate between XRD result and HR-TEM.

Thank you for the comment. Yes, XRD and TEM are correlated. However, I do not have the facility to perform TEM; therefore, I performed only SEM. I think the characterization presented in the manuscript suffices to identify the modification of MMT.

5 The application of modified clay should be studied.

Thank you for the suggestion. I am currently studying the application of modified clay.

Reviewer 3 Report

This manuscript does not meet the minimum standards of a scientific publication. The language is poor. The science is poor. The authors hsould study the "Handbook of Clay Science" and then write their paper. Organically modified montmorillonite (Mt) has been described in detail and I do not see the progress in science and/or knowledge in this paper. If the crude sample contains 52 wt% Mt, the authors must provide the other minerals and/or amorphous oxides present. There are many uncomprehensible items such as '1) 30% slurry (in water?); (2) what is a slurry with organic intercalation? (3) what is the standard solution of methylene blue? (4) what is sodification temperature? (5) carbonyl peak should be carbonate peak and so on.

Author Response

e is from a local mineral hill and contains sand grains. The sodification process not only modifies the MMT but also removes its impurities.

There are many uncomprehensible items such as

'1) 30% slurry (in water?);

Thank you for the comment. Are you referring to section 2.3? The 30% refers to the amount of STAC with respect to the amount of Na-MMT. The text has been revised to clarify this.

  • what is a slurry with organic intercalation?

Please refer to section 2.3, which specifies “organic intercalation agent, STAC.”

  • what is the standard solution of methylene blue?

I followed the national standard of China, GB/T 20973-2020. If necessary, I can add the specifies the standard.

  • what is sodification temperature?

The sodification temperature is one of the conditions. Please see Figure 3.

(5) carbonyl peak should be carbonate peak and so on.

Thank you for the comment. The relevant text has been revised according to your suggestion.

Reviewer 4 Report

The entire manuscript must be proofread because of typing and language errors. The innovation of the work needs to be better described. In the methodology part, referencing, including the equations used, must be included. There is only a description of the results and not a deep discussion to understand the material, not making it suitable for publication in this renowned Journal.

Author Response

The entire manuscript must be proofread because of typing and language errors. The innovation of the work needs to be better described. In the methodology part, referencing, including the equations used, must be included. There is only a description of the results and not a deep discussion to understand the material, not making it suitable for publication in this renowned Journal.

The manuscript describes the raw materials that were obtained from the local mineral hill and that have been used by the company for over 10 years. For better development of the company, it is developing organic bentonite for drilling fluid. Considering that the composition of bentonite from different sources varies considerably, the organically modified experimental methods have been used in numerous studies, although the results differ completely.

I have added the references in the methodology part.

Reviewer 5 Report

The present study shows very interesting data. Firstly, raw Ca-montmorillonite is converted to Na-MMT by optimizing the influence of the content of Na2CO3, reaction time, and stirring speed on MMT content, rotary viscosity and colloid index. Afterward, the optimal Na-MMT was modified using organic ammonium chloride salt. The paper presents a lot of valuable information but still needs improvements, as sown below. The references list should be widened.

Some parts need higher clarity and precision, as follows:

-          The title could be changed to “Preparation and Properties of Organically Modified Na-Montmorillonite”.

-          Abstract should more precisely state what was done in the study. Which conditions were used while producing organically modified MMT?  In which way was it optimized? What does “good properties” of Na-MMT mean? The abstract seems too short, results of the instrumental analysis could be shortly explained.

-          How can MMT be produced from calcium??? Improve the text.

-          In which way was the preparation of Na-MMT chosen (Section 2.2.)? Is it from the literature?

-          Comment on the change in rotational viscosity and its practical meaning in Section 3.1.

-          FTIR analysis results should be appropriately done. To which groups do the obtained bands correspond? Do peaks obtained at about 3450 and 1640 cm-1 correspond to montmorillonite? In which war are their sizes affected by the quantity of this mineral? It seems the clay mineral quantity is not gradually changed with the increased organic salt amount. Compare the results with XRD. Consult the literature and refer to it. What may be the cause of a wide the most intensive band at about 1000 cm-1? What is the size of that band influenced by? For this discussion, you can refer to https://doi.org/10.1016/j.clay.2022.106410.

-          Why is d increased in organically modified MMT? Refer to the literature. Discuss also other peaks obtained by the XRD.

-          Compare the obtained TGA/DTG results to more literature data. Do the peaks obtained at 255 C show the disintegration of free octadecyl trimethyl ammonium chloride?

-          Some technical remarks:

-           Use superscripts (F- and Cr2+).

-          There is no need to form abbreviations for the terms mentioned several times only. Please check the whole text.

-          Use always the right terms. Change MMT to Na-MMT wherever it is appropriate.

-          You should mention the figure first and later discuss the results, and then, after that text, place the corresponding image.

Author Response

- The title could be changed to “Preparation and Properties of Organically Modified Na-Montmorillonite”.

Thank you for the comment. The relevant text has been revised.

- Abstract should more precisely state what was done in the study. Which conditions were used while producing organically modified MMT?  In which way was it optimized? What does “good properties” of Na-MMT mean? The abstract seems too short, results of the instrumental analysis could be shortly explained.

Thank you for the comment. The relevant text has been revised.

-  How can MMT be produced from calcium??? Improve the text.

Raw MMT in China is essentially Ca-MMT.

- In which way was the preparation of Na-MMT chosen (Section 2.2.)? Is it from the literature?

The preparation of Na-MMT was based on company experience.

-  Comment on the change in rotational viscosity and its practical meaning in Section 3.1.

Thank you for the comment. I have commented on the meaning of rotational viscosity with the help of Ref. [22]. In practice, it is a metric of the shear resistance of a system such as a drilling fluid.

- FTIR analysis results should be appropriately done. To which groups do the obtained bands correspond? Do peaks obtained at about 3450 and 1640 cm-1 correspond to montmorillonite?

Thank you for the comment. The FTIR results are thoroughly discussed in the revised manuscript. The absorption bands you refer to originate from water in the interlayer.

In which way are their sizes affected by the quantity of this mineral?

Thank you for the comment. The water in the sample can be characterized via TGA. Other methods to detect water in the sample include TGA and GC.

It seems the clay mineral quantity is not gradually changed with the increased

organic salt amount. Compare the results with XRD.

Thank you for the comment. Unfortunately, I do not understand the meaning of your comment. The XRD spectra characterize the change in the MMT, notably in the range 20°–27°. The results reveal a small amount of zeolite and quartz.

Consult the literature and refer to it.What may be the cause of a wide the most intensive band at about 1000 cm-1? What is the size of that band influenced by? For this discussion, you can refer to https://doi.org/10.1016/j.clay.2022.106410.

Thanks for providing a reference. As mentioned in the literature, “These results do not coincide with the amount of amorphous material determined by XRF and XRD, most likely because in such heterogeneous materials, other bands of crystalline quartz should also be taken into account, especially the largest one at about 1000 cm1. There is also a possibility that the heterogeneity of the samples and the small portion of the material taken for XRD analyses are the reason for the different results.” Thus, numerous absorptions are possible at 1000 cm1; therefore, it is not possible to differentiate them. This article is cited in the text to explain the lack of discussion of the band at 1000 cm−1.

- Why is d increased in organically modified MMT? Refer to the literature. Discuss also other peaks obtained by the XRD.

Thank you for the comment. The relevant text has been revised.

- Compare the obtained TGA/DTG results to more literature data. Do the peaks obtained at 255 C show the disintegration of free octadecyl trimethyl ammonium chloride?

Thank you for the comment. The boiling point of free octadecyl trimethyl ammonium chloride (STAC) (CAS: 112-03-8) is 249 °C. TGA and DTG reveal several peaks near 250 °C, which correspond to the surface, interlayer, and inner of MMT. 

-  Some technical remarks:

 -  Use superscripts (F- and Cr2+).

Thank you for the comment. The text has been revised.

- There is no need to form abbreviations for the terms mentioned several times only. Please check the whole text.

Thank you for the comment. The text has been revised.

- Use always the right terms. Change MMT to Na-MMT wherever it is appropriate.

Thank you for the comment. The text has been revised.

- You should mention the figure first and later discuss the results, and then, after that text, place the corresponding image.
Thank you for the comment. The text has been revised.

Round 2

Reviewer 1 Report

I should thank the authors for giving time and effort to respond carefully to my comments and suggestions. I checked the authors’ responses and the manuscript, and I think everything is now fine. I believe the manuscript is ready for publication in Materials.

Author Response

Thanks a lot!

Reviewer 3 Report

The paper has been improved but still needs another revision.

1; The authors mention methylene blue adsorption, but they do not report results of these experiments.

2. I do not understand what is "colloidal rate". According to formula 2 the colloidal rate is lower when the free volume V in the formula increases. I would have expected the opposite.

3 The abbreviation of montmorillonite is Mt. Avoid MMT

4. 2 lines above figure 6: intensities of Si-O ... vibrations significantly enhanced, indicating that Mt was well modified: what do the authors mean?

5. General remark: Organic modification of Mt is well known for many years. There is nothing new in the paper except for the viscosity measurements and colloidal rates. These aspects should be the core of the paper, but it is not. This is a pity.

Author Response

The paper has been improved but still needs another revision.

  1. 1.The authors mention methylene blue adsorption, but they do not report results of these experiments.

Methylene blue adsorption is directly correlated to the bentonite content in montmorillonite, and the corresponding data are presented in Fig. 1–4.

  1. I do not understand what is "colloidal rate". According to formula 2 the colloidal rate is lower when the free volume V in the formula increases. I would have expected the opposite.

Colloidal rate is the complete swelling of bentonite in the solvent. Within a certain time, it will be suspended to a degree in the solvent. For example, the data in the below picture are 43.2, 52.4, 93.3, and 95.8. The higher the value, the easier it will be to disperse bentonite in the solvent.

3 The abbreviation of montmorillonite is Mt. Avoid MMT

This abbreviation has been used in many literature reports; hence, we used it in our paper. Further, we think MMT is more appropriate.

4.2 lines above figure 6: intensities of Si-O ... vibrations significantly enhanced, indicating that Mt was well modified: what do the authors mean?

This line has been revised for clarity.

  1. General remark: Organic modification of Mt is well known for many years. There is nothing new in the paper except for the viscosity measurements and colloidal rates. These aspects should be the core of the paper, but it is not. This is a pity.

Yes, you are right. However, bentonite has different properties due to its different origins. Especially for a company, product performance directly determines its sales. The results in this paper are different from those in other similar papers.

Reviewer 4 Report

Although the authors have made suggestions of the other reviewers, I maintain the previous opinion.

Author Response

Thanks for your review all the time.

Reviewer 5 Report

The manuscript has been improved, but some important points are still missing, so it should be revised again before publication.

-       Newly added results on the chemical identification of montmorillonites are strange. First, there are no important oxides that are essential to describe the clayey soil, there is no recalculation using separately done loss on ignition after firing at 1000 °C. This way presented; chemical analysis does not reveal enough information.

The comments from the last revision need to be clarified (and inserted in the text, as follows:

-  How can MMT be produced from calcium??? Improve the text.

Raw MMT in China is essentially Ca-MMT.

This is the sentence in the manuscript: “In China, MMT is mainly obtained from calcium [1,2].”

The claim is not true, you cannot produce clay minerals from Ca. Please, improve the text, as said in the first revision.

- In which way was the preparation of Na-MMT chosen (Section 2.2.)? Is it from the literature?

The preparation of Na-MMT was based on company experience.

This should be then stated in the text.

-  To which groups do the obtained bands correspond? Do peaks obtained at about 3450 and 1640 cm-1 correspond to montmorillonite?

Thank you for the comment. The FTIR results are thoroughly discussed in the revised manuscript. The absorption bands you refer to originate from water in the interlayer.

In which way are their sizes affected by the quantity of this mineral?

Thank you for the comment. The water in the sample can be characterized via TGA. Other methods to detect water in the sample include TGA and GC.

Consult the literature and refer to it. What may be the cause of a wide the most intensive band at about 1000 cm-1? What is the size of that band influenced by? For this discussion, you can refer to https://doi.org/10.1016/j.clay.2022.106410.

Thanks for providing a reference. As mentioned in the literature, “These results do not coincide with the amount of amorphous material determined by XRF and XRD, most likely because in such heterogeneous materials, other bands of crystalline quartz should also be taken into account, especially the largest one at about 1000 cm1. There is also a possibility that the heterogeneity of the samples and the small portion of the material taken for XRD analyses are the reason for the different results.” Thus, numerous absorptions are possible at 1000 cm1; therefore, it is not possible to differentiate them. This article is cited in the text to explain the lack of discussion of the band at 1000 cm−1.

This point is not improved in the manuscript, and the explanation about the sizes and shifts of FTIR bands can be found in the mentioned paper in the first revision. Discuss this in your example.

It seems the clay mineral quantity is not gradually changed with the increased

organic salt amount. Compare the results with XRD.

Thank you for the comment. Unfortunately, I do not understand the meaning of your comment. The XRD spectra characterize the change in the MMT, notably in the range 20°–27°. The results reveal a small amount of zeolite and quartz.

While discussing the XRD results, all of the peaks should be defined. To which mineral are the peaks about 3-5 2Θ related? What are the peaks' sizes informing us considering the quantities?

- Compare the obtained TGA/DTG results to more literature data. Do the peaks obtained at 255 C show the disintegration of free octadecyl trimethyl ammonium chloride?

Thank you for the comment. The boiling point of free octadecyl trimethyl ammonium chloride (STAC) (CAS: 112-03-8) is 249 °C. TGA and DTG reveal several peaks near 250 °C, which correspond to the surface, interlayer, and inner of MMT.

This should be stated in the text. The reviewers` comments are usually those that need to be inserted in the text for clarity and improvement of conclusions. What are wide peaks in DSC at about 650 °C from? Every peak obtained in all the instrumental analyses should be discussed and more literature used to find the explanations.

-       The “calling for” tables and figures is still after these results appear in the text, which should be vice-versa as stated in the first revision.

Author Response

The manuscript has been improved, but some important points are still missing, so it should be revised again before publication.

 -       Newly added results on the chemical identification of montmorillonites are strange. First, there are no important oxides that are essential to describe the clayey soil, there is no recalculation using separately done loss on ignition after firing at 1000 °C. This way presented; chemical analysis does not reveal enough information.

I think the table 1 is the best way to compare the differences before and after sodification. As the reviewer said, chemical analysis does not reveal enough information. Yes, I only reveal the information of Sodium element.

The comments from the last revision need to be clarified (and inserted in the text, as follows:

 -  How can MMT be produced from calcium??? Improve the text.

Raw MMT in China is essentially Ca-MMT.

This is the sentence in the manuscript: “In China, MMT is mainly obtained from calcium [1,2].”

The claim is not true, you cannot produce clay minerals from Ca. Please, improve the text, as said in the first revision.

This sentence has been revised for clarity.

- In which way was the preparation of Na-MMT chosen (Section 2.2.)? Is it from the literature?

The preparation of Na-MMT was based on company experience.

This should be then stated in the text.

This has been added to the revised manuscript.

-  To which groups do the obtained bands correspond? Do peaks obtained at about 3450 and 1640 cm-1 correspond to montmorillonite?

Thank you for the comment. The FTIR results are thoroughly discussed in the revised manuscript. The absorption bands you refer to originate from water in the interlayer.

In which way are their sizes affected by the quantity of this mineral?

Thank you for the comment. The water in the sample can be characterized via TGA. Other methods to detect water in the sample include TGA and GC.

Consult the literature and refer to it. What may be the cause of a wide the most intensive band at about 1000 cm-1? What is the size of that band influenced by? For this discussion, you can refer to https://doi.org/10.1016/j.clay.2022.106410.

Thanks for providing a reference. As mentioned in the literature, “These results do not coincide with the amount of amorphous material determined by XRF and XRD, most likely because in such heterogeneous materials, other bands of crystalline quartz should also be taken into account, especially the largest one at about 1000 cm−1. There is also a possibility that the heterogeneity of the samples and the small portion of the material taken for XRD analyses are the reason for the different results.” Thus, numerous absorptions are possible at 1000 cm−1; therefore, it is not possible to differentiate them. This article is cited in the text to explain the lack of discussion of the band at 1000 cm−1.

This point is not improved in the manuscript, and the explanation about the sizes and shifts of FTIR bands can be found in the mentioned paper in the first revision. Discuss this in your example.

The manuscript has been revised to include this point. However, this article (https://doi.org/10.1016/j.clay.2022.106410,) doesn’t mention the MMT.

It seems the clay mineral quantity is not gradually changed with the increased

organic salt amount. Compare the results with XRD.

Thank you for the comment. Unfortunately, I do not understand the meaning of your comment. The XRD spectra characterize the change in the MMT, notably in the range of 20°–27°. The results reveal a small amount of zeolite and quartz.

While discussing the XRD results, all of the peaks should be defined. To which mineral are the peaks about 3-5 2Θ related? What are the peaks' sizes informing us considering the quantities?

Thanks for your suggestion. The focus of our study was on the degree of organic intercalation. Impurity removal is incidental.

 - Compare the obtained TGA/DTG results to more literature data. Do the peaks obtained at 255 C show the disintegration of free octadecyl trimethyl ammonium chloride?

Thank you for the comment. The boiling point of free octadecyl trimethyl ammonium chloride (STAC) (CAS: 112-03-8) is 249 °C. TGA and DTG reveal several peaks near 250 °C, which correspond to the surface, interlayer, and inner of MMT.

This should be stated in the text. The reviewers` comments are usually those that need to be inserted in the text for clarity and improvement of conclusions. What are wide peaks in DSC at about 650 °C from? Every peak obtained in all the instrumental analyses should be discussed and more literature used to find the explanations.

This point has been included in the revised manuscript.

-       The “calling for” tables and figures is still after these results appear in the text, which should be vice-versa as stated in the first revision.

   This has been appropriately modified in the revised text.

   We are thankful to the reviewers for their detailed review. Some of the reviewer's questions are very professional and in-depth. However, we did not modify it as required because the focus of the paper was organic modification, and other phenomena were considered to be analyzed in future.

Round 3

Reviewer 3 Report

This paper is improved. SOme points need to be clarified before publication.

1. According to the handbook of clay science the abbrevaition of montmorillonite is Mt, not MMT

2. The authors use methylene blue adsorption, but there are no data in the section results. The results of the methylene blue adsorption must be presented.

Author Response

1. According to the handbook of clay science the abbrevaition of montmorillonite is Mt, not MMT

Thanks for your attention. But I find many examples using MMT as the abbrevaition of montmorillonite in many journals such as https://doi.org/10.1016/j.clay.2023.106887, https://doi.org/10.1016/j.saa.2022.121289, and https://doi.org/10.1016/j.porgcoat.2022.106782.

2. The authors use methylene blue adsorption, but there are no data in the section results. The results of the methylene blue adsorption must be presented.

Thank your for review in detail. We can use the equation (1) to figure out the results of the methylene blue adsorption. So I don't think it's necessary to include process data in the article.

Reviewer 5 Report

The paper has been improved and can be published.

Author Response

Thanks a lot!